# Biochemical Fulvic Acid Modification for Phosphate Crystal Inhibition in Water and Fertilizer Integration

**DOI:** 10.3390/ma15031174

**Published:** 2022-02-03

**Authors:** Jianyun Li, Zihan Nie, Zhao Fan, Chunguang Li, Bingbing Liu, Quanxian Hua, Cuihong Hou

**Affiliations:** 1School of Civil Engineering and Architecture, Zhengzhou University of Aeronautics, Zhengzhou 450046, China; zualijy@126.com (J.L.); lichunguang@zua.edu.cn (C.L.); 2School of Chemical Engineering, National Center for Research and Popularization on Calcium, Magnesium, Phosphate & Compound Fertilizer Technology, Zhengzhou University, Zhengzhou 450001, China; liubingbing@zzu.edu.cn (B.L.); huaqx@zzu.edu.cn (Q.H.); hch92@zzu.edu.cn (C.H.); 3School of Environmental Economics, Henan Finance University, Zhengzhou 450046, China; henanzhaoyuan@163.com

**Keywords:** biochemical fulvic acid, acrylic acid, modification, water, fertilizer integration

## Abstract

Biochemical fulvic acid (BFA), produced by organic wastes composting, is the complex organic matter with various functional groups. A novel modified biochemical fulvic acid (MBFA) which possessed stronger chelating ability had been synthesized by the grafting copolymerization of BFA and acrylic acid (AA). Results showed that MBFA effectively inhibited the crystallization of calcium phosphate and increased the concentration of phosphate in water solution. The optimum reaction conditions optimized by Box–Behnken design and response surface methodology were reaction temperature 69.24 °C, the mass of monomer to fulvic acid ratio 0.713, the initiator dosage 19.78%, and phosphate crystal-inhibition extent was 96.89%. IR spectra demonstrated AA was grafted onto BFA. XRD data and SEM images appeared the formation and growth of calcium phosphate crystals was effectively inhibited by MBFA.

## 1. Introduction

Grain production mainly depends on yield and acreage. Fertilizer is the most important element for enhancing crop yield. Given that China is the most populous country in the world, in order to meet the food demand for the country, fertilization and irrigation have been the main ways to achieve high-yield grain production [1,2,3,4,5]. Despite the increasing use of organic fertilizers, the great majority of fertilizers are still based on mineral resources, especially phosphate compound fertilizers. Excessive application of chemical fertilizers easily brings serious environmental problems, e.g., nutrient loss, soil quality deterioration, and water eutrophication [6,7,8]. On the other hand, China is facing the problem of water shortage [2,5,9,10,11]. Water and fertilizer integration can realize the simultaneous supply of nutrients and water. Its main advantage is the establishment of optimal water and nutritive regime directly in the plant root system [12,13], which significantly increases the uptake of nutrients such as nitrogen and phosphate by plant roots and effectively improves the water and fertilizer utilization efficiency [14,15]. Drip irrigation technologies also bring new problems in nutrient loss and water waste. When natural water is used to dissolve water soluble fertilizer containing phosphate, some insoluble substances are easily formed, and the deposition causes the blockage of drip irrigation system [16]. The clogging has seriously restricted the promotion of water and fertilizer integration. Research shows that chemical clogging could be relieved by enhancing the acidity of irrigation water or using acidified fertilizers [17,18]. Due to its strong acidizing effect, hydrochloric or nitric acid has also been considered as an effective additive for eliminating the phenomena [19,20]. However, both chemical acidification and long-term application of acidic fertilizers will cause harm to crop and soil, as well as corrosion of drip irrigation equipment.

Humic substance, derived from plant remains and their decomposition products, is widely found in water or soil. It is composed of the main element such as C, H, O, N, S, and the difference in the content of main elements is determined by the various sources [21]. It contains many oxygenated function groups, such as hydroxyl and carboxyl groups, which have strong chelation and adsorption [22]. Results showed that the adsorption of humic substance to calcium ions could effectively improve the solubility of calcium phosphate [23,24]. It behaves as weak acid polyelectrolytes and the occurrence of anionic charged sites, and accounts for the ability to retain cations [25]. The chelate is most stable when a five or six-membered ring with metal cation is formed [26]. Therefore, the higher content of carboxyl in humic substance indicates the better chelation and adsorption effect and the more obvious improvement of the solubility of calcium phosphate. Due to its remarkable chelating property, humic substance is considered as crystal inhibitors in water and fertilizer integration. According to the molecular weight and solubility in water, this substance can be divided into humin, humic acid, and fulvic acid [27,28]. The last has smaller molar mass, higher solubility, higher carboxylic contents than the others, and it has stronger complexation [26]. In addition, fulvic aid is also a kind of soil conditioner, which can improve soil quality and promote the growth of crops [29,30].

Obtained through microbial fermentation, biochemical humic substance is a kind of weak acidic organic substance whose chemical property is like that of natural humic substance [31]. This is attributed to various functional groups of biochemical humic substance including carboxylic acid. Organic wastes such as crop straws and livestock manure can be used as raw materials for aerobic degradation to produce biochemical humic substance. During anaerobic process, the structure of humic substance varied with time including for a variety of functional groups—such as carboxylic, phenolic and hydroxylic groups [32]. The final product of aerobic degradation is a variety of organic matter mainly composed of biochemical humic acid and biochemical fulvic acid, the latter has a higher carboxyl and phenolic group content, compared with the former [33,34]. As it has higher biological activity and lower price, BFA is thus more suitable as an additive for water and fertilizer integration.

However, BFA has a few disadvantages, which limit its application. There are many impurities in BFA, and its dosage is 5–10 times of mineral fulvic acid in order to achieve the same effect. Another problem is that BFA is highly hydroscopic, and it will lead to fertilizer caking when its content reaches 2.5% in fertilizer compounds.

Studies indicate that cation binding sites in fulvic acid are mainly carboxylic groups [35,36,37], and the chemical modification of BFA to improve its carboxyl content can effectively solve the problem. Among the various modifiers, AA is chosen because it is easy to polymerize and able to provide the necessary carboxylic groups [38,39,40,41]. Potassium peroxydisulfate is a quite effective initiator in grafting copolymerization [42,43,44]. So, in the paper, the grafting copolymerization experiments were designed for improving the chelating and adsorption capacity of BFA, in which AA was used as modifier and potassium peroxydisulfate as initiator.

## 2. Materials and Methods

### 2.1. Purification of BFA

BFA used in the experiments was supplied by a certain company in China, the content of C, H, O, N, S and ash was 46.32%, 3.26%, 45.12%, 1.26%, 3.58%, 0.46% using Elementar Analysensysteme GmbH (vario MICRO, ELEMENTAR, Hanau, Hessen, GER), and the micropollutant of Cu, Pb, Cr was not detected using ICP-AES (iCAP6500, Thermo, Waltham, MA, USA). The raw material fulvic acid needed to be further purified before the experiment began, due to some insoluble impurities, soluble amino acids, carbohydrates, etc. All other chemicals used in the experiments were of analytical purity and used without any further purification.

The acid-insoluble humin and humic acid were removed by base-dissolving acidification method. The water-soluble impurities, such as amino acids and polysaccharides, were removed by ethanol. BFA content was determined by potassium dichromic oxidation, as well as the COOH groups of it measured by calcium acetate method.

### 2.2. Modification of BFA

BFA and potassium hydroxide were dissolved in deionized water at a proportion of 2% and 0.4% (*w/v*), respectively. The amount of potassium persulfate and AA was 5–25%, 50–150% based on the amount of biochemical fulvic, respectively, and dissolved in a small amount of deionized water. The BFA solution was heated to a set temperature. Then, potassium persulfate and AA solutions were added into BFA solution with a speed of 4 μL/s by a peristaltic pump for 2400 s. After the reaction was completed, the solution was taken out and the water-insoluble matter filtered out. The supernatant was extracted with equal volume of anhydrous ethanol for 3 times. The precipitate was collected and sent to be drying at 50 °C, the dried product was ground into powder, which was MBFA [45,46]. All experiments were arranged with single factor test and further optimized by response surface method design.

### 2.3. Determination of Crystal-Inhibition Performance of MBFA

#### 2.3.1. Crystal-Inhibition Experiments

A certain mass of MBFA was accurately weighed and dissolved in deionized water. Then, the CaCl_2_ solution (0.04 mol/L, 10 mL) and the K_2_HPO_4_ solution (0.04 mol/L, 5 mL) were added into MBFA solution and the volume fixed to 100 mL (1 g/L in borax buffer). Being stood for 0.5 h, the filtrate was collected by centrifugation and filtration. After that, the content of phosphorus in the filtrate was determined following the vandomolybdate yellow colorimetric method by using UV-Vis spectrophotometer. The crystal-inhibition extent of MBFA was obtained according to the following formula:(1)$$\eta =\frac{C_{1}-C_{CK}}{C_{0}-C_{CK}}\times 100\%$$
where *η* is the crystal-inhibition extent, %; *C*_1_ is the phosphorus content in the solution after adding BFA (P_2_O_5_, the same below), mg/L; *C_CK_* is the phosphorus content in the solution without BFA, mg/L; *C*_0_ is the initial phosphorus content, mg/L.

#### 2.3.2. Particle Size Experiment

The mixture composed of CaCl_2_ and K_2_HPO_4_ solution (1 g/L in borax buffer) was added into the stirring tank of the laser particle size analyzer (WJL-628, INESA, Shanghai, China). The parameters of the laser particle size analyzer were set as ultrasonic time of 180 s, stirring rate of 1800 r/h and circulating flow rate of 1200 r/h. BFA solution was quickly poured into the stirred tank and the timing was started. The granularity of solid particles in suspension was measured and recorded at set intervals. The dispersibility of BFA to calcium phosphate can be analyzed by this method.

### 2.4. Characterization and Analysis

Fourier transform infrared spectrum (Nicolet iS10, Nicolet, Waltham, MA, USA) was used to investigate the changes of functional groups of BFA. The infrared wave number range was 500~4000 cm^−1^. X-ray powder diffractometer (XRD-6000, Shimadzu, Japan) was used to analyze the changes of calcium phosphate crystal structure peaks after adding BFA. The scanning speed was 10° (2θ)/min, and the 2θ scanning range was 10–110°. Scanning electron microscope (Auriga FIB, Zeiss, Germany) was used to analyze the surface morphology data of calcium phosphate crystals before and after adding BFA.

## 3. Results and Discussions

### 3.1. Single Factor Experiments

With the change of the level of each factor, the change trends of the crystal-inhibition extent were similar, which first increased and then decreased. It revealed that the optimum reaction conditions were as following, reaction time of 1 h, initiator dosage of 20 wt %, reaction temperature of 70 °C, the ratio of monomer to BFA of 0.75. The results of single factor experiment were shown as Figure 1, and the data of 0.2 to 1.0 g/L represent the different concentration of the amount of MBFA in solution.

### 3.2. Response Surface Model Analysis

According to the results of single-factor experiments, the crystal-inhibition extent was chosen as the response value (*Y*). Reaction temperature (*X*_1_), monomers ratio (*X*_2_) and initiator dosage (*X*_3_) were the experimental factors. Box-Behnken was used to design a three-factor three-level experiment for the data model. The control factor levels for those experimental parameters were shown as Table 1.

Box–Behnken design presented a total of 17 experimental runs, which has been conducted to investigate the effects of various factors on *Y* [47]. The experimental conditions and results of each group were shown as follows in Table 2.

#### 3.2.1. Analysis of Variance of Regression Equation Model and Factors

All factors affecting the crystal-inhibition performance were non-linear. Through analysis of variance of the equation, the results are shown in Table 3.

From the table of ANOVA, the Model F-value of 113.56 and *p*-value less than 0.0001 implied that the model was significant. It indicated that the error between the actual value and the equation was small, and the relationship between the response value and each factor can be fairly reflected. The “Lack of Fit *F*-value” of 4.01 implied the Lack of Fit was not significant relative to the pure error, which indicated good fitting of the suggested model. The difference between “Predicted *R^2^*” of 0.9157 and “Adjusted *R^2^*” of 0.9845 was less than 0.2. This meant that the predictive value will be reasonable. The “Adequate Precision” ratio of 28.126 indicated an adequate signal. This model can be used to study the crystal-inhibition extent of MBFA on calcium phosphate.

The “*p*-values” less than 0.0500 indicated that model terms were significant. As seen in Table 3, temperature (*X_1_*) and monomers ratio (*X_2_*) were statistically significant, suggesting that the two factors significantly affected crystal-inhibition ability of modified chemical fulvic acid. It also showed that the quadratic term coefficients (*X_1_^2^*, *X_2_^2^* and *X_3_^2^*) and interaction coefficient (*X_1_X_2_*) were significant model terms [48]. The other term coefficients had no significant effect on crystal-inhibition ability of MBFA. The order of influence of three factors on crystal-inhibition ability of MBFA was as follows: monomers ratio (*X_2_*) > reaction temperature (*X_1_*)> initiator dosage (*X_3_*). According to the above analysis, the reduced actual equation (Eq. 2) that just used the significant terms was fitted.
*Y* = −199.97 + 7.67*X_1_* + 44.94*X_2_* + 0.26*X_1_X_2_* − 0.06*X_1_^2^* − 44.52*X_2_^2^* − 0.02*X_3_^2^*(2)

#### 3.2.2. Interaction

The relationships between the operational parameters were presented by plotting two independent variables with the crystal-inhibition extent using a three-dimensional contour map. The variation regulation of Y under the effects of two independent variables is shown as Figure 2.

As shown in Figure 2a, the crystal-inhibition extent increased with increasing temperature in a certain monomers ratio, but further increase of the temperature resulted in a reverse trend. The variation trend of crystal-inhibition extent with monomers ratio was similar to that mentioned above. This trend was also found in Figure 2b,c. However, it was obvious that the radian of the surface in Figure 2b,c was significantly smaller than that in Figure 2a. According to the above analysis, combined with Figure 2d–f, it could be determined that the order of the influence of interaction on the crystal-inhibition performance of MBFA was: reaction temperature / monomers ratio (*X_1_X_2_*) > reaction temperature / initiator dosage (*X_1_X_3_*) > monomers ratio / initiator dosage (*X_2_X_3_*).

From the optimized mathematical model, the optimal levels of the three factors were: reaction temperature of 69.24 °C, monomers ratio of 0.713, initiator dosage of 19.78%, and the corresponding response was 97.10%. Finally, verification experiments were performed using the optimized culture conditions and the experimentally observed crystal-inhibition extent was 96.89%, which is in good agreement with the model prediction of 97.10%.

**Figure 2 materials-15-01174-f002:**
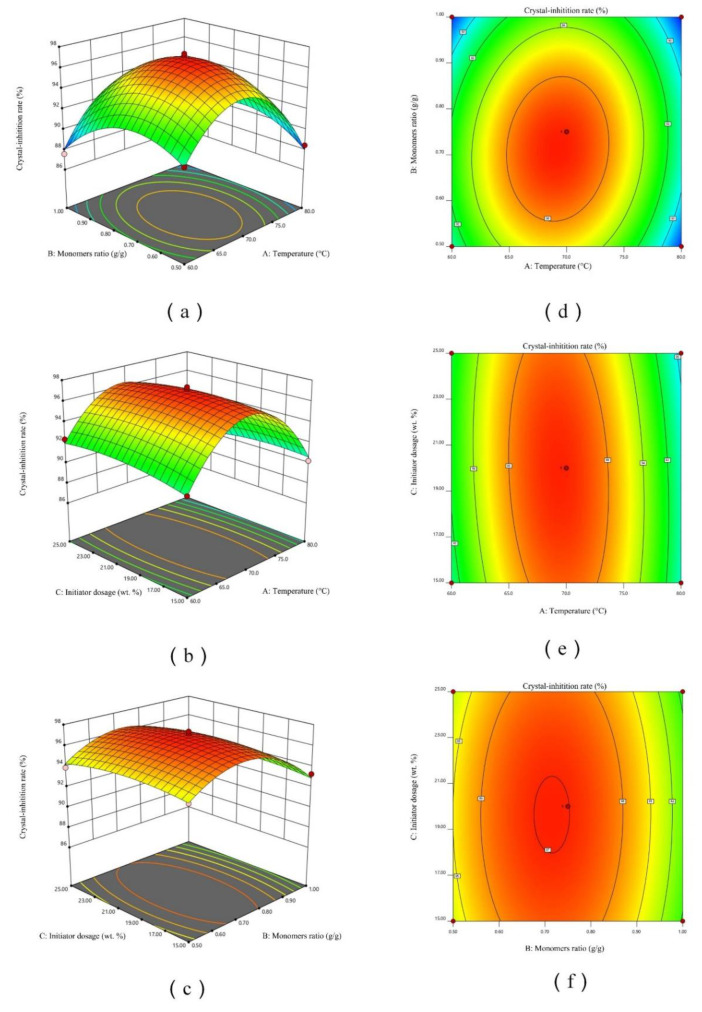
3D response surface plots showing the interaction between (**a**) temperature and monomers ratio; (**b**) temperature and initiator dosage; (**c**) monomers ratio and initiator dosage. Contour plots showing the interaction between (**d**) temperature and monomers ratio; (**e**) temperature and initiator dosage; (**f**) monomers ratio and initiator dosage. The 3rd factor in each figure is at the middle level.

### 3.3. Effect of MBFA on Particle-Size Distribution of Calcium Phosphate

MBFA could disperse phosphate suspension and sediments in solution. So, adding MBFA into calcium phosphate suspension could effectively reduce the size of solid particles. As shown in Figure 3, with the increase of time, the particle size of solid decreased gradually. After about 60 s, the particle size of half the number of particles had been reduced to 1 μm. It indicated that MBFA had a significant and efficient dispersion effect on calcium phosphate crystals.

### 3.4. FT-IR Analyses

FTIR spectra could be used to detect the presence of characteristic functional groups in organic compounds. It was utilized in our study to corroborate the occurrence of grafting copolymerization. The comparison of IR spectra of AA, BFA and MBFA is shown as Figure 4, where appeared clear differences from BFA and the modified one. After grafting copolymerization of AA onto BFA, a new stronger peak was detected at 1106 cm^−1^ which was attributed to -C-O- stretching. This peak was weak in BFA spectrum. The broadening of the characteristic peak of hydroxyl, 2800–3800 cm^−1^, also confirmed the increase of oxygenated groups. The results indicated that the grafting reaction between AA and BFA was successful. The peaks detected at 1632 and 973 cm^−1^ were indicative of C=C and =C-H group, respectively. The intensity of these two absorption peaks decreased obviously in MBFA spectrum. This highlighted the fact that more AA was grafted onto BFA in the form of polymer.

### 3.5. XRD Analyses

X-ray diffraction analysis contributed to the internal molecular structure information of materials, which was used to determine the crystal structure. Figure 5 shows the X-ray diffraction contrast patterns of calcium phosphate, MBFA and calcium phosphate added with MBFA. In curve (a), there were obvious diffraction peaks at 30.94°, 44.86°, 55.87° and 74.37°, among which the peaks were sharp at the first two. It indicated that calcium phosphate crystals grew well. In curve (c), the diffraction peaks of 30.94°, 44.86°, 55.87° and 74.37° were greatly weakened, and the diffraction peaks of 34.71° and 50.46° were enhanced. The original diffraction peak disappeared at 83.89° and an amorphous peak appeared at 28.93°. It showed that the growth of calcium phosphate crystals was inhibited, and the crystal form of calcium phosphate changed. As a result, other amorphous structures were formed, and the crystallinity became poor.

### 3.6. SEM Imaging

The morphology of the crystals was observed by scanning electron microscope. SEM images of calcium phosphate crystals are shown as Figure 6. Calcium phosphate without MBFA was tightly bound with large in size and regular morphology. After adding MBFA solution at the concentration of 0.4 g/L, the morphology of calcium phosphate changed to many solid particles with loose irregular structure, which reduced the adhesion between particles. As a result the addition of MBFA could inhibit crystal growth and made it form fine and irregular microcrystals. MBFA attached to surface of microcrystals could increase the solubility of calcium phosphate in water, thus making it difficult to form dense calcium phosphate precipitates.

## 4. Conclusions

Grafting polymerization of AA onto BFA formed a novel copolymer with stronger metal-chelating capacity, MBFA. This was attributed to more oxygenated function groups existing in BFA. The crystal-inhibition extent of MBFA on calcium phosphate reached 96.89% under optimized conditions which was the reaction temperature of 69.24 °C, the mass of monomer to fulvic acid ratio of 0.713, the initiator dosage of 19.78%. IR spectrum indicated that grafting reaction between AA and BFA was successful. Compared with BFA, the content of carboxyl groups in MBFA was significantly increased. X-ray diffraction diagram showed that the growth of calcium phosphate crystals was inhibited, the crystal form of calcium phosphate changed. Scanning electron microscopy confirmed that the crystal structure of calcium phosphate changed from dense to loose after adding MBFA. MBFA could inhibit the growth of calcium phosphate crystals and make it produce irregular microcrystals.

## Figures and Tables

**Figure 1 materials-15-01174-f001:**
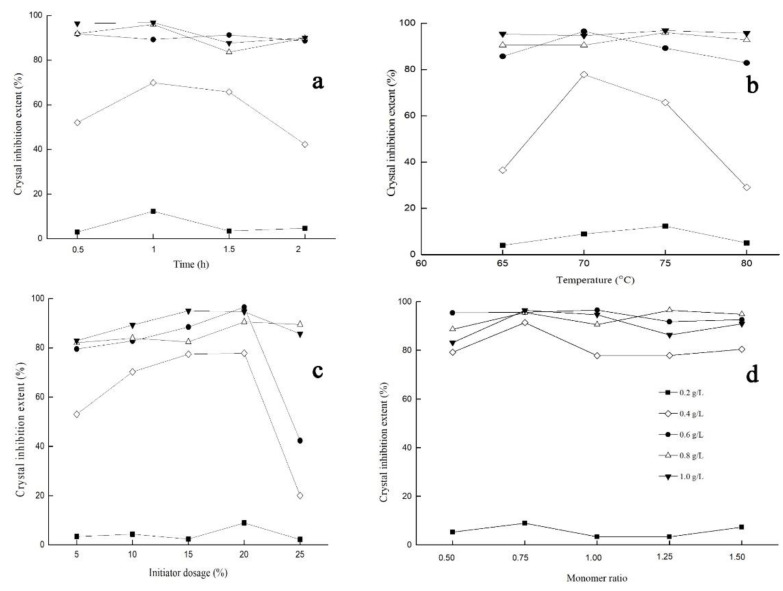
Effects of certain factors on MBFA. (**a**) time, (**b**) temperature, (**c**) initiator dosage, (**d**) monoter ratio.

**Figure 3 materials-15-01174-f003:**
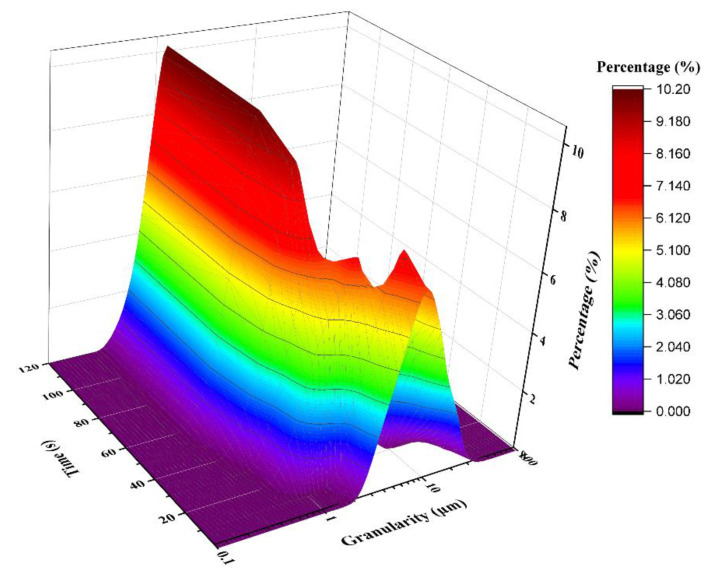
Particle size distribution of calcium phosphate crystal at different time.

**Figure 4 materials-15-01174-f004:**
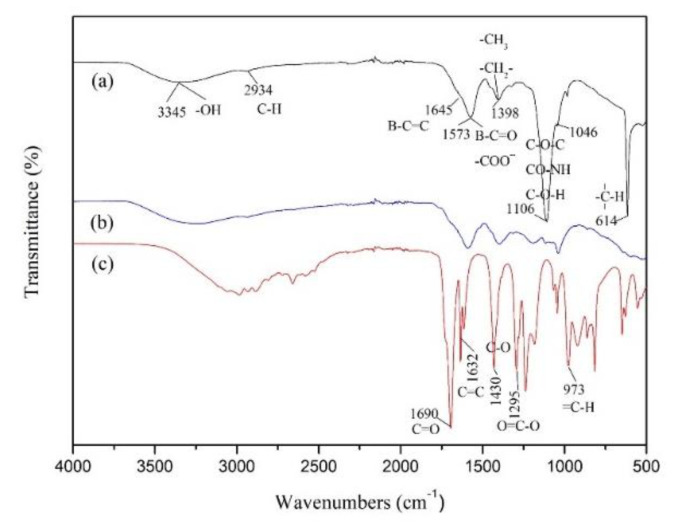
IR spectra of MBFA (**a**), BFA (**b**) and AA (**c**).

**Figure 5 materials-15-01174-f005:**
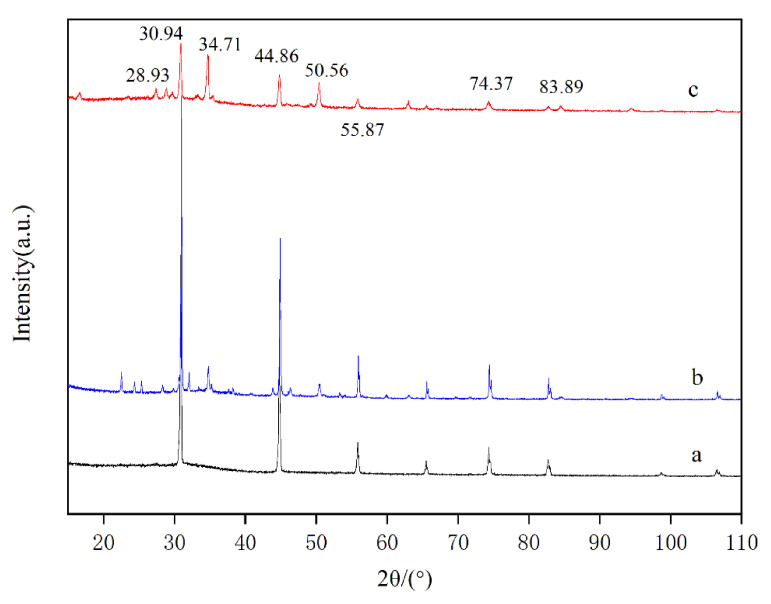
XRD patterns of calcium phosphate (**a**), MBFA (**b**), calcium phosphate with MBFA (**c**).

**Figure 6 materials-15-01174-f006:**
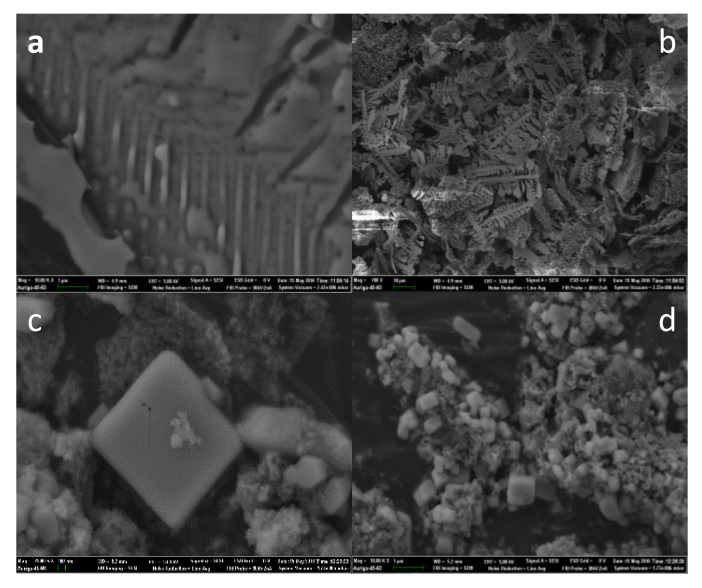
SEM images of calcium phosphate crystals (**a**,**b**), crystals with MBFA (**c**,**d**).

**Table 1 materials-15-01174-t001:** Levels of independent variables.

Parameters	Codes	Levels		
		−1	0	1
temperature/°C	*X_1_*	60	70	80
monomer ratio	*X_2_*	0.50	0.75	1.00
initiator dosage/*wt* %	*X_3_*	15	20	25

**Table 2 materials-15-01174-t002:** Experimental design with response of independent variables.

Run No.	*X_1_* (°C)	*X_2_* (g/g)	*X_3_* (*wt* %)	Exp-*Y* (%)	Pre-*Y* (%)
1	70	0.75	20.00	97.14	97.01
2	80	0.75	15.00	90.24	90.62
3	60	0.75	25.00	92.36	91.98
4	80	1.00	20.00	87.79	87.80
5	70	0.50	15.00	94.47	94.51
6	70	0.50	25.00	93.91	94.30
7	80	0.50	20.00	88.46	88.04
8	80	0.75	25.00	89.54	89.57
9	70	0.75	20.00	97.35	97.01
10	60	0.50	20.00	90.88	90.87
11	70	0.75	20.00	96.59	97.01
12	70	1.00	25.00	92.82	92.78
13	70	0.75	20.00	96.92	97.01
14	60	0.75	15.00	91.29	91.26
15	70	0.75	20.00	97.05	97.01
16	70	1.00	15.00	93.30	92.91
17	60	1.00	20.00	87.58	88.00

**Table 3 materials-15-01174-t003:** Analysis of variance (ANOVA) for response surface quadratic mode.

Source	Sum of Squares	*d_f_*	Mean Square	*F*-Value	*p*-Value	Significant
Model	186.46	9	20.72	113.56	<0.0001	Significant
*X_1_*	4.62	1	4.62	25.33	0.0015	
*X_2_*	4.85	1	4.85	26.59	0.0013	
*X_3_*	0.0561	1	0.0561	0.3076	0.5964	
*X_1_X_2_*	1.73	1	1.73	9.48	0.0178	
*X_1_X_3_*	0.7832	1	0.7832	4.29	0.0770	
*X_2_X_3_*	0.0016	1	0.0016	0.0088	0.9280	
*X_1_^2^*	129.69	1	129.69	710.89	<0.0001	
*X_2_^2^*	32.60	1	32.60	178.69	<0.0001	
*X_3_^2^*	1.53	1	1.53	8.38	0.0232	
**Residual**	1.28	7	0.1824			
Lack of Fit	0.9585	3	0.3195	4.01	0.1065	Not significant
Pure Error	0.3186	4	0.0797			
**Cor Total**	187.74	16				
** *R* ^2^ **	0.9932		**Predicted *R*^2^**		0.9157	
**Adjusted** ** *R* ** ** ^2^ **	0.9845		**Adeq Precision**		28.1256	

## Data Availability

Some or all data that support the findings of this study are available from the corresponding author upon reasonable request.

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
