# Peer review of "Biochemical Fulvic Acid Modification for Phosphate Crystal Inhibition in Water and Fertilizer Integration"

_materials, 2022, doi:10.3390/ma15031174_

Round 1

Reviewer 1 Report

The subject of the work is biochemical fulvic acid, produced during the composting of organic waste, which is a complex organic matter with various functional groups and is modified for Phosphate Crystal Inhibition in water and fertilizer integration.

There are significant gaps in the data that need to be supplemented in the paper

1). The study lacks the elemental composition of the biochemical humic used, obtained as a result of microbial fermentation and the content of inorganic micropollutants (ash).

2). The authors write: A certain amount of potassium persulfate and acrylic acid were dissolved in a small amount of deionized water, respectively. The biochemical fulvic acid solution was heated to a set temperature. The amount of dosed reagents or the patent number should be given. The conditions are imprecise.

The research is interesting and original, the work is written correctly, the text is clear and easy to read, the layout is logical and the conclusions are correct. The work after supplementing can be published.

Author Response

1.1 The study lacks the elemental composition of the biochemical humic used, obtained as a result of microbial fermentation and the content of inorganic micropollutants (ash).

Response: We feel sorry that we did not provide enough information. Based on your comments, the composition data have been added.

Revised: ……, the content of C, H, O, N, S and ash was 46.32%, 3.26%, 45.12%, 1.26%, 3.58%, 0.46% using Elementar Analysensysteme GmbH (vario MICRO), and the micropollutant of Cu, Pb, Cr was not detected using ICP-AES. ……

1.2 The authors write: A certain amount of potassium persulfate and acrylic acid were dissolved in a small amount of deionized water, respectively. The biochemical fulvic acid solution was heated to a set temperature. The amount of dosed reagents or the patent number should be given. The conditions are imprecise.

Response: Thank you for your valuable suggestions. According to your opinion, we supplemented the details of single factor experiments.

Revised: ……, The amount of potassium persulfate and acrylic acid was 5-25%, 50-150% based on the amount of biochemical fulvic, respectively, and dissolved in a small amount of deionized water. ……

Reviewer 2 Report

Li et al presented a paper devoted to the application of artificial humic-like mixture modified with polyacrylic acid in order to inhibit the growth of phosphate crystals, which aggravate the draining systems. Despite the general importance of the topic, I am not satisfied with the paper. In its current state the manuscript and the work is not of high quality. Most disappointing is the absence of experimental details, which made it impossible to evaluate the procedures and results. And some experimental details are frustrating, e.g. extraction of water supernatant with methanol or, in general, methanol extraction method without any citation whatsoever. Second, the results of graft-polymerization through fulvic acids is described poorly. Only three FT-IR spectra (without explanation of experimental set-up) were provided. And I have doubts about the results. How does the mixture of the monomer and FA in the particular proportions look like before the reaction? I don't see a strong band of C=O in the modified sample in Fig. 4. Where are all the COOH? In the spectrum on-acidic fragments appeared , which don't help chelation. What happens with FA under oxidative conditions without any monomer? Finally, I have doubts about co-polymerization. How can author be sure, that this wasn't s mixture of two independent products? Additionally, the discussion of the results is quite poor. Authors present some statistical outcomes without clear explanation. Moreover, abbreviation from tables are not explained, e.g. in Table 2 Exp-Y and Pre-Y. Last, but not least, I don’t understand the title with its “biochemical FA modification”. FAs are always products of biochemical reactions. And modification was purely chemical – free-radical polymerization initiated by persulfate. My other comments are presented below. I strongly recommend authors to resubmit the manuscript with some significant improvements, and I hope my comments will help to correct it.

1) The paper author cited is devoted to the Cl2 impact on biofilms. And I don't see how Cl2 may help with insoluble salts, such as phosphates.

2) The Intro part has to be rearranged for the clarity and the language has to be improved. 

3) Humic substances, humic acids, fulvic acids - plural. They are not single molecules

4) Biochemical FAs. Model, synthetic, artificial but definitely not biochemical HS

5) What was the reason to exclude humic acids fraction from the mixture, which underwent reaction, if they are perfectly dissolved in alkali solution?

6) To which monomer concentration are figs. 1a-c corresponding?

7) Fig.5 with XRD results is missing. Currently, it is a replica of Fig. 4.

Author Response

2.1 The paper author cited is devoted to the Cl2 impact on biofilms. And I don't see how Cl2 may help with insoluble salts, such as phosphates.

Response: Thanks for your careful reviews. We have corrected the wrong statement and References. Chemical clogging was relieved by increasing H+ instead of Cl2 in water and fertilizer integration, and hydrochloric or nitric acid was better additive.

Revised: ……, Researches show that chemical clogging could be relieved by enhancing the acidity of irrigation water or using acidified fertilizers [17,18]. Due to its strong acidizing effect, hydrochloric or nitric acid has also been considered as an effective additive for eliminating the phenomena [19,20]. ……

2.2 The Intro part has to be rearranged for the clarity and the language has to be improved.

Response: Thank you for your nice opinion on our manuscript. We have improved this part according to your suggestion.

2.3 Humic substances, humic acids, fulvic acids - plural. They are not single molecules.

Response: Thank you for reviewing our article. Please allow us to answer this question. Humic substance can be divided into humin, humic acid and fulvic acid according to the solubility, each partition is multiple composition complex and has not single molecule.

2.4 Biochemical FAs. Model, synthetic, artificial but definitely not biochemical HS

Response: Thanks for your careful reviews. We think you are correct. But biochemical fulvic acid was extracted from biochemical humic substance, or, biochemical fulivic acid is one of the components of biochemical humic substance.

2.5 What was the reason to exclude humic acids fraction from the mixture, which underwent reaction, if they are perfectly dissolved in alkali solution?

Response: Thanks for your nice comments on our manuscript. Humic acid can be dissolved in alkali solution, but not in acidic solution, fulvic acid can be dissolved in both. The application of fulvic acid is more broadcast than that of humic acid for decreasing the chemical clogging in water and fertilizer integration.

2.6 To which monomer concentration are figs. 1a-c corresponding?

Response: Thanks for your careful reviews. Please allow us to answer your question in detail. The four curves are consistent in Figs. 1a-d, and the figure legend is arranged in Fig. 1d.

2.7 Fig.5 with XRD results is missing. Currently, it is a replica of Fig. 4.

Response: Thanks for your careful checks. Fig.5 with XRD results is added.

Round 2

Reviewer 1 Report

The work has been corrected according to the reviewer's guidelines. I have no more comments. The work can be published.

Author Response

no more comments

Reviewer 2 Report

Authors tried to improve the manuscript. They managed to decrease the size of crystals. However, I was surprised that my major comments on the structural investigation of the product were ignored and only minor comments got response. Consequently, I am repeating my comments but in a more explicit numbered way.

1) What is a methanol extraction method? Did author mean that the mixture of FA was extracted with methanol?

2) How was the concentration of phosphorus determined by UV? What was the protocol, what complex did authors use?

3) The results of graft-polymerization are described poorly. Authors presented 3 FT-IR spectra from which I couldn’t deduce that the reaction was a success. Spectrum of acrylic acid includes strong band of C=O from COOH-group. But it was missing from the spectrum of the reaction product. How was the increase in COOH-group content proved?

4) The problem with FT-IR part is that the mixture of two independent products and the co-polymer may possess similar bands. Polymerization of acrylic acid is spontaneous, and authors correctly highlighted that the C=C bonds disappeared and C-H band was quite strong. However, this would happen without FA anyway. Authors extracted the reaction yield by methanol, but acrylic acid oligomers and even polyacrylic acid are also soluble in methanol. Therefore, FT-IR doesn’t prove the successful co-polymerization. In my opinion in order to prove the reaction the correlation NMR (e.g. HSQC) is required and, even better, reaction should have been conducted with 13C-labeled acrylic acid. This is out of scope of this research but then, the conclusions about success of the co-polymerization should be also toned down.

5) Authors should have provided the FT-IR spectra of the blank experiments: polymerization of acrylic acid and polymerization of FA independently.

6) I agree with the authors that this particular FA product is not conventional FA from soil or peat. However, Biochemical humic substances sounds strange. I suggest renaming it. E.g. biotechnologically produced fulvic acid – the name I found in the literature (http://dx.doi.org/10.17582/journal.pjz/2019.51.3.961.970). Also, the abbreviation is required. There is no need to keep the full name through the text. For example, Biotechnologically produced fulvic acid can be designated as BPFA.

Author Response

  1. What is a methanol extraction method? Did author mean that the mixture of FA was extracted with methanol.

Response: Thanks for your careful reviews. In the paper’s experiment, methanol was used to remove the water-soluble ingredient and extract the supernatant for removing the water in the precipitate (biochemical fulvic acid) and preventing biochemical fulvic acid agglomerating during oven drying. In fact, anhydrous ethanol was used instead of methanol in the following experiments of biochemical fulvic acid, and we revise ‘methanol’ to ‘ethanol’ in the paper.

Revised:

  • •••••, The water-soluble impurities, such as amino acids and polysaccharides, were removed by ethanol. ••••••
  • •••••, The supernatant was extracted with equal volume of anhydrous ethanol for 3 times. ••••••
  1. How was the concentration of phosphorus determined by UV? What was the protocol, what complex did authors use?

Response: Thanks for your careful reviews. The original expression is not detailed, and we have replenished the method for determining phosphorus.

The concentration of phosphorus was determined following the vandomolybdate yellow colorimetric method by using UV-Vis spectrophotometer.

Revised: ••••••, the content of phosphorus in the filtrate was determined following the vandomolybdate yellow colorimetric method by using UV-Vis spectrophotometer. ••••••

  1. The results of graft-polymerization are described poorly. Authors presented 3 FT-IR spectra from which I couldn’t deduce that the reaction was a success. Spectrum of acrylic acid includes strong band of C=O from COOH-group. But it was missing from the spectrum of the reaction product. How was the increase in COOH-group content proved?

       4) The problem with FT-IR part is that the mixture of two independent products and the co-polymer may possess similar bands. Polymerization of acrylic acid is spontaneous, and authors correctly highlighted that the C=C bonds disappeared and C-H band was quite strong. However, this would happen without FA anyway. Authors extracted the reaction yield by methanol, but acrylic acid oligomers and even polyacrylic acid are also soluble in methanol. Therefore, FT-IR doesn’t prove the successful co-polymerization. In my opinion in order to prove the reaction the correlation NMR (e.g. HSQC) is required and, even better, reaction should have been conducted with 13C-labeled acrylic acid. This is out of scope of this research but then, the conclusions about success of the co-polymerization should be also toned down.

       5) Authors should have provided the FT-IR spectra of the blank experiments: polymerization of acrylic acid and polymerization of FA independently.

Response:

Because Question 3 to Question 5 belong to FT-IR, please allow me to answer them together.

Reviewer is an expert with extensive practice at FT-IR spectral analysis, but authors of the paper are poor at spectral reading, we are good at laboratory tests. In the paper, all tests were organized by single factor design and respond surface method design, all results were shown by the data obtained with the tests, spectral analysis and micromorphology were supplementary proofs, including IR, XRD, NMR and SEM.

Preparation of modified biochemical fulvic acid was based on biochemical fulvic acid which was purified by H+ and OH- solution. Modified biochemical fulvic acid was purified with water, methanol or ethanol, acrylic acid or its polymer did not exist in modified biochemical fulvic acid.

6) I agree with the authors that this particular FA product is not conventional FA from soil or peat. However, Biochemical humic substances sounds strange. I suggest renaming it. E.g. biotechnologically produced fulvic acid – the name I found in the literature (http://dx.doi.org/10.17582/journal.pjz/2019.51.3.961.970). Also, the abbreviation is required. There is no need to keep the full name through the text. For example, Biotechnologically produced fulvic acid can be designated as BPFA.

Response: Thanks for your constructive suggestion. Biochemical fulvic acid, acrylic acid, modified biochemical fulvic acid was designated as BFA, AA, MBFA, respectively. The literature (http://dx.doi.org/10.17582/journal.pjz/2019.51.3.961.970) was also entered in the References.
